# Evaluating AI Courses: A Valid and Reliable Instrument for Assessing Artificial-Intelligence Learning through Comparative Self-Assessment

Matthias Carl Laupichler [1,*], Alexandra Aster [1], Jan-Ole Perschewski [2] and Johannes Schleiss [2]

1   Institute of Medical Education, University Hospital Bonn, 53127 Bonn, Germany; alexandra.aster@ukbonn.de
2   Artificial Intelligence Lab, Otto von Guericke University Magdeburg, 39106 Magdeburg, Germany;
    jan-ole.perschewski@ovgu.de (J.-O.P.); johannes.schleiss@ovgu.de (J.S.)
*   Correspondence: matthias.laupichler@ukbonn.de

**Abstract:** A growing number of courses seek to increase the basic artificial-intelligence skills ("AI literacy") of their participants. At this time, there is no valid and reliable measurement tool that can be used to assess AI-learning gains. However, the existence of such a tool would be important to enable quality assurance and comparability. In this study, a validated AI-literacy-assessment instrument, the "scale for the assessment of non-experts' AI literacy" (SNAIL) was adapted and used to evaluate an undergraduate AI course. We investigated whether the scale can be used to reliably evaluate AI courses and whether mediator variables, such as attitudes toward AI or participation in other AI courses, had an influence on learning gains. In addition to the traditional mean comparisons (i.e., *t*-tests), the comparative self-assessment (CSA) gain was calculated, which allowed for a more meaningful assessment of the increase in AI literacy. We found preliminary evidence that the adapted SNAIL questionnaire enables a valid evaluation of AI-learning gains. In particular, distinctions among different subconstructs and the differentiation constructs, such as attitudes toward AI, seem to be possible with the help of the SNAIL questionnaire.

**Keywords:** AI literacy; AI-literacy scale; artificial intelligence education; assessment; course evaluation; comparative self-assessment





## 1. Introduction

### 1.1. AI Literacy

Artificial intelligence (AI) is permeating more and more areas of daily life. While no universal definition of AI exists, most definitions agree that it is "a branch of computer science dealing with the simulation of intelligent behavior in computers" [1] and that AI represents the idea of "computer programs that have some of the qualities of the human mind" [2]. Examples of AI use can be found in a wide variety of fields, including mundane applications such as movie recommendations [3] and video games [4] and applications in highly specialized professions such as medicine [5] and engineering [6,7]. In the course of such developments, more and more people are coming into contact with AI applications, consciously or unconsciously. In order to be able to deal with AI in a meaningful and outcome-oriented way and to be able to assess possible benefits as well as risks, a certain understanding of AI is essential. This basic understanding, which most, or even all, individuals should have, is often referred to as AI literacy.

While different definitions of AI literacy exist, the most commonly cited definition comes from a paper written by Long and Magerko [8]. They described AI literacy as "a set of competencies that enables individuals to critically evaluate AI technologies; communicate and collaborate effectively with AI; and use AI as a tool online, at home, and in the workplace" [8], (p. 2).

Various educational projects aim to improve individuals' AI literacy. Introductory courses on AI are offered at a wide variety of educational levels, starting as early as kindergarten and elementary school [9,10], continuing through K-12 education [11–13], and ending with higher education and adult education in universities and similar institutions [14,15]. To measure the impact and effectiveness of these educational projects, some of them have been examined in evaluation studies and accompanying research, the results of which have been published.

Many researchers resort to self-created or unvalidated instruments to measure learning success. Other researchers do not measure learning success at all, but limit their findings to the lowest level of the Kirkpatrick model [16] by reporting direct, affective reactions toward a course. However, assessing learning gains with a reliable, objective, and valid instrument is important in uncovering potential problems in the delivery of AI literacy content and in evaluating the quality of AI courses. For this reason, many researchers have called for the development of AI-literacy-assessment instruments of high psychometric quality [15,17,18].

### 1.2. Assessing AI Literacy

Several relatively well-validated scales already exist that try to capture affective attitudes toward AI [19–21]. However, the measurement of AI literacy that meet psychometric quality standards is still a fairly new field of research. In fact, existing measurement tools are still evolving and an optimal assessment tool has not yet been established. Nevertheless, some promising initial efforts have been made to develop AI-literacy-assessment instruments.

The first AI literacy scale was published by [22] and included four sub-factors of AI literacy: "awareness", "usage", "evaluation", and "ethics". The authors of that study drew on previous research in digital literacy and human–AI interaction to develop their scale. Another study that reported the creation of a set of AI-literacy items was published by Pinski and Benlian [23]. They presented the findings of a preliminary study that distributed the items to 50 individuals. The resulting dataset was then used to draw conclusions about the structure of AI competencies, using structural-equation modeling.

An unpublished manuscript by Carolus et al. [24], describing the development of their scale for AI-literacy assessment, has not yet undergone a peer-review process. Nonetheless, this scale represents an interesting contribution to AI-literacy assessment, as they used a top-down approach and based their item development on the categories introduced in the well-cited review by Ng et al. [25].

In contrast, Laupichler et al. [26] followed a bottom-up approach and used a Delphi study to generate a set of content-valid items that were relevant and representative for the field of AI literacy. Those items were validated in another study, which is currently undergoing a peer-review process. The items generated in the Delphi study were presented to a sample of more than 400 participants and, subsequently, analyzed through an exploratory factor analysis that found three AI-literacy factors: technical understanding, critical appraisal, and practical application [27].

### 1.3. Using an AI-Literacy-Assessment Instrument to Evaluate Learning Gains

While the aforementioned instruments have been validated using appropriate samples, it has not yet been determined whether they are suitable for assessing learning gains or evaluating the effectiveness of AI courses. However, a valid and reliable evaluation of AI courses is essential for several reasons. First, high-quality assessment tools enable quality-assurance procedures to be implemented. Second, such evaluation tools could be used to identify potential strengths and weaknesses of an AI course in a resource-efficient manner. The information obtained could be used as part of the continuous and iterative improvement of a course. Third, evaluating different courses with the same instrument would allow comparability across individual courses or study programs. Such

comparisons could then be used, for example, for external evaluation of course offerings or for program specialization.

We hypothesize that AI-literacy scales developed to assess the status quo of individuals' AI knowledge can also be used to evaluate the quality of AI courses, with some minor adaptations. It is particularly important that these AI-course-evaluation instruments be validated for this purpose in order to obtain meaningful and comparable results. In this study, the "scale for the assessment of non-experts' AI literacy" (SNAIL), which was developed by Laupichler et al. [27], was used, because it was validated on a sufficiently large sample and the items can be adapted particularly well for course evaluation. However, one of the other scales described above ([22–24]) could just as easily have been used, as the adaptations described below can be applied to all items, regardless of their origin.

The original test instrument was changed only with respect to two parameters, the adoption of the language (optional) and the introduction of self-assessment of differences via a "retrospective assessment" and a "post-assessment". Concerning the first parameter, the original scale by Laupichler et al. [28] was originally validated in English and the course participants were German native speakers. Therefore, the original items were first translated into German (see the Materials and Methods section of this article). This prevented misunderstandings and lowered the cognitive barrier to completing the evaluation questionnaire, which in turn had a positive effect on the response rate.

The second modification of the original scale allows for the measurement of differences in self-assessed AI literacy that may occur by attending the AI course. For this purpose, each item was presented as a retrospective assessment version and a post-assessment version, meaning that the participants had to assess their individual competency on every item, respectively, in a retrospective manner (i.e., looking back to the time before the course) and with respect to their current capability (i.e., after taking part in the course). The retrospective/post method is often more suitable for assessing learning gains than the traditional pre/post test (one assessment before and one after a course, [28]) because it is subject to fewer biases. Especially when assessing skills prior to educational intervention, learners tend to overestimate their competency because they often cannot yet fully grasp the depth of the field, an effect commonly referred to as response-shift bias [29,30].

### 1.4. Research Questions

The objective of this study was to investigate the applicability of the validated AI literacy scale for the evaluation of AI courses. Specifically, we aimed to assess the scale's effectiveness in measuring changes in learning gains through comparative self-assessment. Additionally, we sought to investigate whether certain items or factors within the scale showed more significant increases in knowledge and skill than others. These distinctions could prove to be valuable in identifying any weaknesses in the evaluated courses, thus facilitating targeted improvements. Therefore, our first research question was:

RQ1: Can the adapted "scale for the assessment of non-experts' AI literacy" be used to reliably and validly assess the learning gains of AI courses?

In addition, we aimed to explore the extent to which course participants' AI literacy was influenced by their attitudes toward AI, and vice versa. If AI literacy and attitudes toward AI are correlated, then it might be advisable to assess attitudes toward AI in future AI-course evaluations. Moreover, if the relationship between the two variables is causal (rather than merely correlative), it might be necessary to take steps to improve participants' attitudes in addition to their learning gains. Thus, our second research question was:

RQ2: Are AI course participants' self-assessed AI literacy and their attitudes toward AI correlated?

Finally, we wanted to investigate the extent to which attending other AI courses prior to the evaluated AI course had an impact on learning success and self-assessment values. Furthermore, AI education does not take place only in formal settings such as courses; students also use other sources, such as educational videos, books, and social media posts,

to learn about AI topics. Therefore, a question on the use of other means of education was added, and its relationship to participants' self-assessments was examined. Accordingly, our third and final research question was:

RQ3: Does AI education outside of an evaluated AI course have an impact on learning gains?

## 2. Materials and Methods

### 2.1. AI Course

The course that was evaluated using the SNAIL questionnaire was an interdisciplinary AI course designed to teach AI skills to undergraduate students. Students from different study programs were allowed to register for the course, which meant that both students who studied computer science or related subjects and students who had relatively little contact with programming and computer science content in their previous studies took part in the course. Although it could be argued that computer science students are experts in the field of AI, it was still reasonable to use the "scale for the assessment of non-experts' AI literacy". Although these students could be expected to have a high technical literacy, they had little or no education that was focused on fostering their AI literacy. Furthermore, the sample did not only consist of computer science students, but also students from other disciplines, so that comparisons between individuals with low and high technology literacy were possible.

The course had a rather technical focus—teaching how artificial neural networks work. It consisted of a lecture, instructor-led exercises, and self-study content and was structured in an application-oriented way. The course scope of all activities amounted to approximately 150 h. The learning outcomes of the course included the application of methods of data analysis with neural networks for solving classification, regression and other statistical problems; the evaluation and application of neural learning techniques for the analysis of complex systems; and the capability of developing neural networks. Thus, the course corresponded mostly to the technical-understanding and practical-application dimensions of the SNAIL questionnaire.

### 2.2. Translation of the "Scale for the Assessment of Non-Experts' AI Literacy" (SNAIL)

To ensure a valid and systematic translation of the SNAIL items, we followed the international recommendations for translating psychological measurement instruments wherever possible [31–33]. Two bilingual speakers whose native language was German independently translated the SNAIL items from English into German. Subsequently, these two translators compared the items and analyzed the differences in the translations in order to reach a common consensus. Thereafter, two additional bilingual speakers (one of whom was a native English speaker) independently translated the items from German back into English. Subsequently, all translators, as well as two methodological experts who were experienced in developing questionnaires, met to identify problems and differences in the scale translation. This expert panel was able to produce a final SNAIL version in German (see Supplementary Material S1).

### 2.3. Evaluation Procedure

We presented each of the 31 SNAIL items and asked the participants about their current self-assessment (after attending the course) and their retrospective self-assessment (at the beginning of the course). Participants were then presented with the five items of the "Attitudes Toward Artificial Intelligence" scale by Sindermann et al. [22]. This attitude scale was used because it is one of the shortest and, thus, one of the most resource-efficient instruments designed to capture attitudes; it had already been validated in the native language of the course participants. In addition, some socio-demographic questions about the participants' ages, fields of study, etc. were collected. Finally, the instrument asked to what extent the participants had already educated themselves on the topic of AI prior to the course, in other courses or with other methods.

*2.4. Data Analysis*

All analyses were performed using Microsoft Excel or IBM SPSS Statistics (version 27). To determine the AI-literacy-learning gains, the mean of the retrospective items was compared to the mean of the post-items, using *t*-tests. The one-tailed *t*-test was used, as it was assumed that students' AI literacy could only improve by attending the course. This was done both at the item level and at the factor level. Since *t*-tests easily become statistically significant, especially in the area of teaching effectiveness and learning-gain evaluation, even for practically irrelevant increases, an additional analysis method was used. The so-called student comparative self-assessment [28,34] is a more valid tool to assess the actual increase in competence or knowledge, as it accounts for the initial level of participants' AI literacy. The calculation of the comparative self-assessment (CSA) gain is described in Raupach et al.'s 2011 article, "Towards outcome-based programme evaluation: using student comparative self-assessments to determine teaching effectiveness" [34], and CSA gain values can range from −100% to +100% (although in reality, negative values are rare). We used Spearman's rank correlation for correlations between metric and ordinal variables (such as AI-literacy-learning gains and the amount of AI education outside of the course) and Pearson correlation for correlations between metric variables. The reliability of the scale was evaluated by assessing the internal consistency (Cronbach's alpha) of the three factors.

One of the items of the technical-understanding factor had to be excluded due to a technical error ("I can describe the concept of explainable AI"), which resulted in a total of 30 items being used to assess AI literacy (13 instead of 14 items in the technical-understanding factor).

# 3. Results

## 3.1. Participants

Because there was no formal enrollment for the course, it was not possible to determine how many people officially attended the course. However, an average of about 40 people attended the lectures and exercises. In total, 25 students (62.5% of all attendees) took part in the study. Study participants were, on average, 22.9 years old (SD = 2.3) and in their sixth semester (M = 6.0, SD = 2.9). More men ($n$ = 16, 64%) than women ($n$ = 9, 36%) participated in the course. As mentioned above, the course was attended by participants from different study programs. Six participants (24%) came from computer science, six (24%) came from a program called "Philosophy Neuroscience Cognition", four (16%) studied statistics, three (12%) studied medical engineering, and two (8%) studied electromobility, computer visualistics, or did not specify their main study program, respectively.

The mean time it took participants to complete the study was 8:01 min (SD = 1:11 min). Participants responded to almost every question and missing values were relatively rare, with an average of 0.2 missing values per respondent (max = 3).

## 3.2. Learning Gains and Reliability

For all 30 AI literacy items used, the mean values of all participants' retrospective assessments were compared to the mean values of all participants' post-assessments, using independent *t*-tests. To test the null hypothesis that the variances were equal, a Levene test was calculated prior to each *t*-test. The homoscedasticity assumption was only violated for one comparison ("I can explain how sensors are used by computers to collect data that can be used for AI purposes"). In this individual case, a Welch test was performed. Based on a significance level of $\alpha$ = 0.05, a significant improvement in performance was found for a majority of the items. Only three items failed to show a statistically significant improvement in the corresponding AI competency: "I can explain the difference between general (or strong) and narrow (or weak) artificial intelligence"; "I can explain why data privacy must be considered when developing and using artificial intelligence applications"; and "I can identify ethical issues surrounding artificial intelligence". The effect size, expressed by Cohen's d, painted a similar picture. Cohen's d was below 0.5 for only five items, indicating

a small effect. All other items had at least a medium effect (d > 0.5), with 14 items showing a strong effect (d > 0.8).

When the analysis was performed at the factor level, similar results were found. There was a significant difference between the retrospective assessments and the post-assessments for all three factors, $t(48) = 4.38$, $p < 0.001$, $d = 1.25$ for the technical-understanding (TU) factor, $t(48) = 3.47$, $p < 0.001$, $d = 1.21$ for the critical-appraisal (CA) factor, and $t(48) = 3.30$, $p < 0.001$, $d = 0.93$ for the practical-application (PA) factor, respectively (see Figure 1).

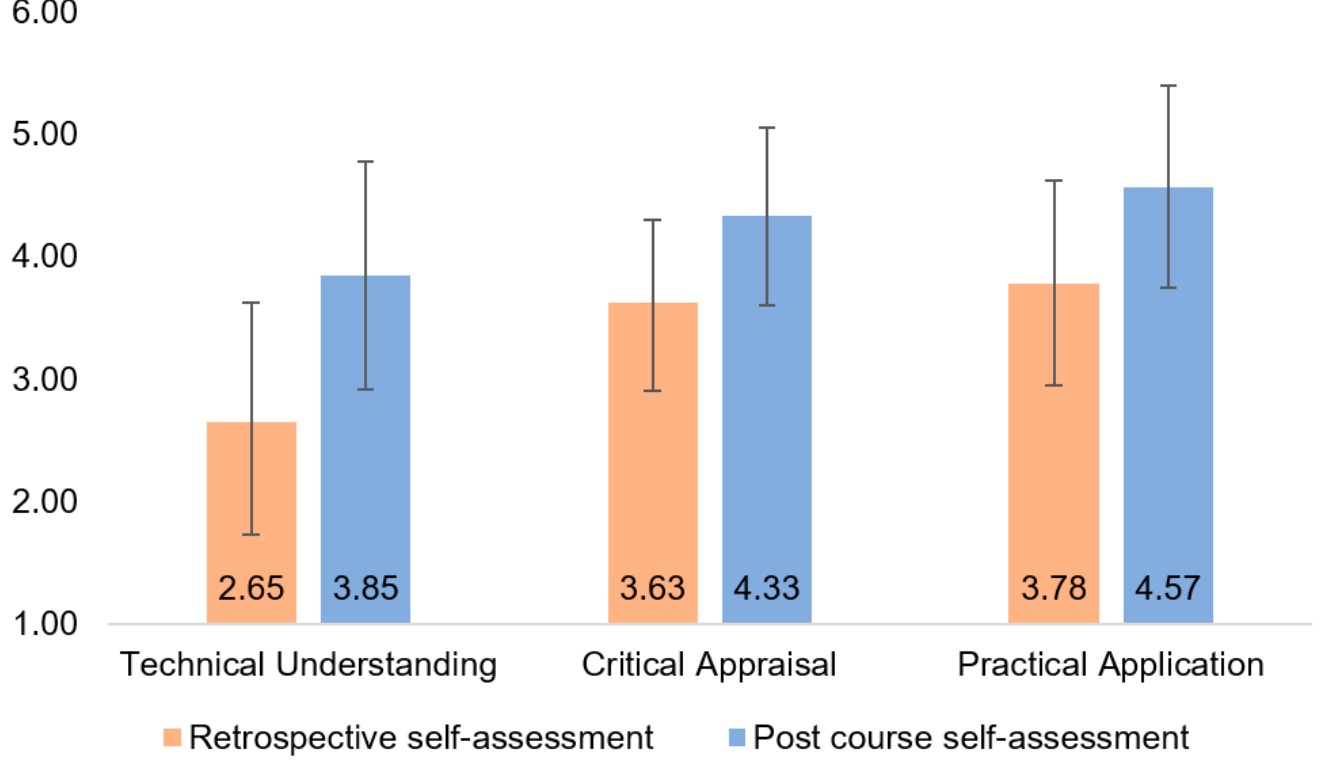

**Figure 1.** Mean values of retrospective assessment and post-self-assessment of all three factors.

As described above, the more informative change parameter CSA gain is reported in the following section. It may be due to the retrospective/post assessments being less susceptible to bias than the traditional pre/post assessments that all CSA gain values were positive, as negative values would imply a loss of AI literacy over the course. However, the actual height of CSA gain varied greatly from item to item (see Figure 2). Some items showed rather small improvements, in the range of 15 to 30%, which could have been due to several reasons. First, students may have already assessed themselves as relatively confident in their relevant competence before attending the course (i.e., the retrospective values were already fairly high). For example, items such as "I can identify ethical issues surrounding artificial intelligence" were rated relatively highly even before attending the course, with an average retrospective-assessment rating of M = 4.20 (SD = 1.02) and a post-assessment rating of M = 4.48 (SD = 0.94) on a six-point Likert scale, leading to a CSA gain of 15.6%. Second, the AI course may have failed to teach the corresponding aspects that are represented by the item. An example of this is the item, "I can describe potential legal problems that may arise when using artificial intelligence", which had an average retrospective-assessment rating of M = 2.88 (SD = 1.14) and a post-assessment rating of M = 3.44 (SD = 1.39). For other items, however, acceptable to good CSA gain was found in the range of 40%, up to more than 50%. A positive example was the item, "I can explain what the term 'artificial neural network' means", for which a CSA gain of 56.8% was found. This was somewhat unsurprising, because one of the main aspects of the course was the teaching of competencies related to artificial neural networks.

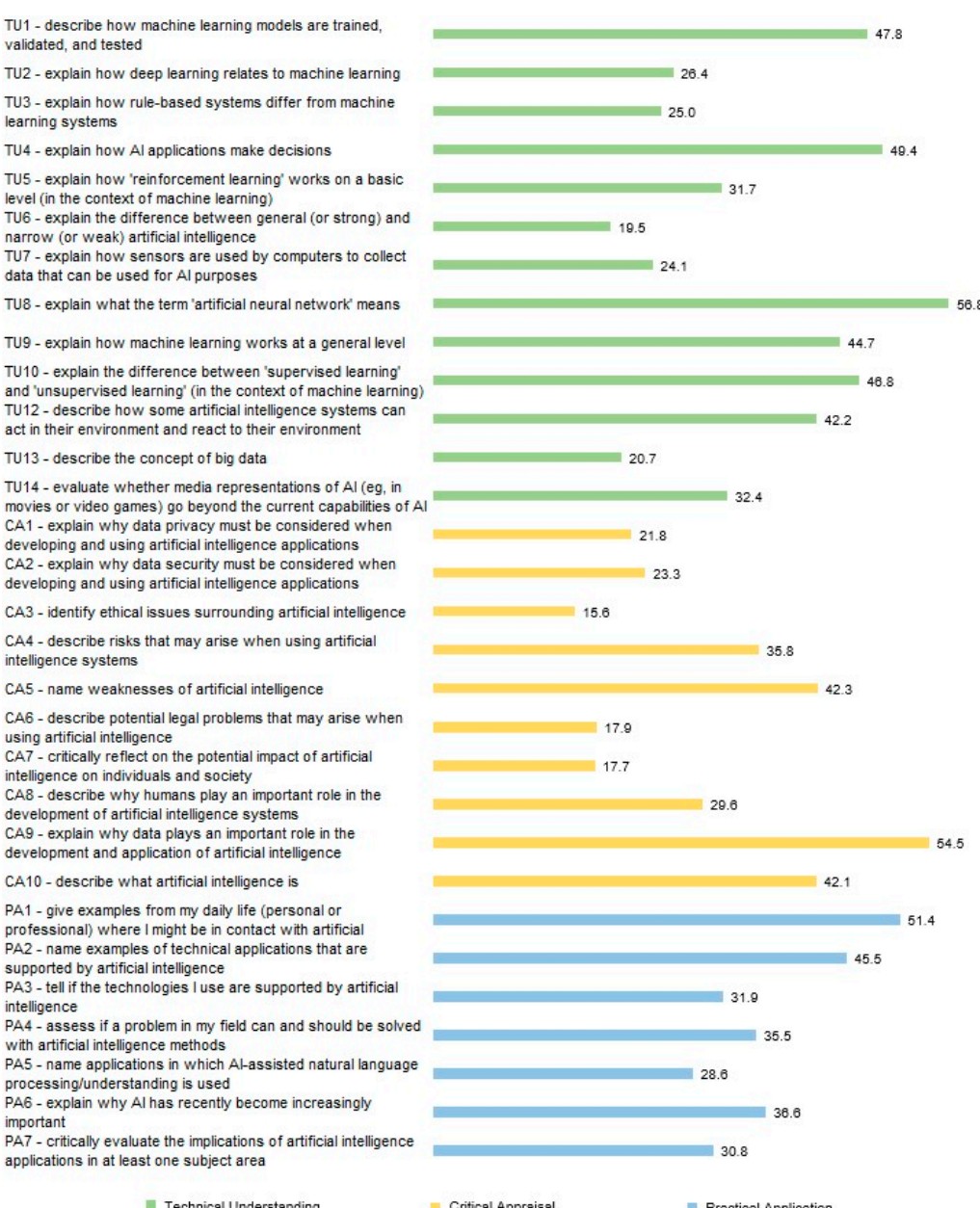

**Figure 2.** Mean CSA gain values, in percentages, for all three factors. Note: CSA gain can reach values from −100% to +100%. Item TU11 was excluded, due to a technical error.

By calculating the CSA gain average over all items of the respective factor, the differences of the CSA gain between the individual items were removed. The CSA gain values were 36.9% for the TU factor, 30.1% for the CA factor, and 37.2% for the PA factor.

The reliability of the individual subscales (i.e., factors) of the SNAIL questionnaire can be rated as good (>0.80) to excellent (>0.90). Cronbach's $\alpha$ for the retrospective assessments was 0.90, 0.92, and 0.85 for the three factors TU, CA, and PA, respectively. The internal consistency of the scales at post-assessment was slightly lower at 0.83 (TU factor), 0.90 (CA factor), and 0.88 (PA factor), but still in the good-to-excellent range.

### 3.3. Relationship between AI Literacy and Attitudes toward AI

The Pearson product–moment correlations between the SNAIL factors (TU, CA, and PA) and the two factors of the "Attitudes Toward Artificial Intelligence" scale, namely "fear" and "acceptance", did not reach statistical significance. In addition, the correlations

between the two "Attitudes Toward Artificial Intelligence" scale factors and the mean scores of the retrospective assessment and the post-assessment of the three SNAIL factors were not significant.

### 3.4. Relationship between AI Education Prior to the Course and Learning Gains

Participants were asked whether and to what extent they had attended other AI courses prior to attending the evaluated course (variable name "other courses"). They were also asked whether and to what extent they had used other means of AI education (such as instructional videos and books.; variable name "other AI education").

Although the correlations between "other courses" and CSA gain were negative for all three factors, this correlation did not reach statistical significance (significance level of $\alpha = 0.5$). Interestingly, the effect was reversed for the variable "other AI education", as all correlations were positive. However, these correlations did not reach statistical significance.

Thereafter, we examined in detail how attending other courses or using other AI educational opportunities affected the absolute retrospective self-assessment and the post-self-assessment. Attending other courses was strongly positively correlated with the assessment scores on the TU factor, with Spearman's $\rho = 0.556$, $p < 0.01$ and Spearman's $\rho = 0.402$, $p = 0.046$ for the retrospective assessment and post-assessment scores, respectively. However, the correlations between "other courses" and the CA or PA factor did not reach statistical significance (see Table 1). Using other means of AI education was also strongly correlated with assessment scores on the TU factor, with Spearman's $\rho = 0.557$, $p < 0.01$ (retrospective assessment) and Spearman's $\rho = 0.684$, $p < 0.001$ (post-assessment). In this case, however, significant positive correlations were also present for the other two factors in the post-assessment (Spearman's $\rho = 0.492$, $p = 0.013$ for the CA factor and Spearman's $\rho = 0.524$, $p < 0.01$ for the PA factor), but not in the retrospective assessment.

**Table 1.** Correlations between CSA gain, retrospective/post assessment for each factor, and the usage of other courses or other AI education.

| Variable | 1 | 2 | 3 | 4 | 5 | 6 | 7 | 8 | 9 | 10 | 11 |
|---|---|---|---|---|---|---|---|---|---|---|---|
| 1. CSA TU | — | | | | | | | | | | |
| 2. CSA CA | 0.382 | — | | | | | | | | | |
| | 0.059 | | | | | | | | | | |
| 3. CSA PA | 0.556 ** | 0.573 ** | — | | | | | | | | |
| | 0.004 | 0.003 | | | | | | | | | |
| 4. TU—retrospective | −0.106 | 0.004 | −0.107 | — | | | | | | | |
| | 0.614 | 0.985 | 0.611 | | | | | | | | |
| 5. TU—post | 0.416 * | 0.385 | 0.321 | 0.678 ** | — | | | | | | |
| | 0.039 | 0.058 | 0.118 | <0.001 | | | | | | | |
| 6. CA—retrospective | −0.254 | −0.234 | −0.169 | 0.357 | 0.271 | — | | | | | |
| | 0.220 | 0.260 | 0.419 | 0.080 | 0.191 | | | | | | |
| 7. CA—post | 0.233 | 0.471 * | 0.415 * | 0.146 | 0.505 * | 0.588 ** | — | | | | |
| | 0.263 | 0.018 | 0.039 | 0.486 | 0.010 | 0.002 | | | | | |
| 8. PA—retrospective | −0.347 | −0.023 | −0.096 | 0.239 | 0.159 | 0.523 ** | 0.266 | — | | | |
| | 0.089 | 0.914 | 0.648 | 0.250 | 0.448 | 0.007 | 0.198 | | | | |
| 9. PA—post | 0.206 | 0.519 ** | 0.593 ** | 0.073 | 0.447 * | 0.302 | 0.729 ** | 0.589 ** | — | | |
| | 0.323 | 0.008 | 0.002 | 0.730 | 0.025 | 0.142 | <0.001 | 0.002 | | | |
| 10. Other courses | −0.111 | −0.348 | −0.244 | 0.556 ** | 0.402 * | 0.364 | 0.066 | 0.043 | −0.147 | — | |
| | 0.597 | 0.088 | 0.239 | 0.004 | 0.046 | 0.074 | 0.755 | 0.838 | 0.482 | | |
| 11. Other AI education | 0.376 | 0.309 | 0.292 | 0.557 ** | 0.684 ** | 0.271 | 0.492 * | 0.332 | 0.524 ** | 0.171 | — |
| | 0.064 | 0.133 | 0.157 | 0.004 | <0.001 | 0.190 | 0.013 | 0.105 | 0.007 | 0.413 | |

\* $p < 0.05$. \*\* $p < 0.01$. Note. TU: technical-understanding factor, CA: critical-appraisal factor, PA: practical-application factor. Retrospective: mean retrospective assessment; post: mean current assessment (after taking the course).

## 4. Discussion

### 4.1. Contextualizing of Results

This study investigated the suitability of Laupichler et al.'s AI-literacy scale, SNAIL [27], for evaluating AI courses. First, evidence was found that suggested that a simple adaptation of the original SNAIL questionnaire allows its use in the context of course evaluations. The

adapted version of SNAIL seems to be able to differentiate between learning objectives and to identify strengths as well as weaknesses of AI courses, providing a balance in evaluating AI-literacy courses. The results indicate that the adapted version of SNAIL is valid and corresponds to the actual AI competencies of course participants. This was supported by several lines of evidence derived from answers to the three research questions.

First, the average learning gain, represented by CSA gain, was particularly pronounced for technical items such as "I can describe how machine learning models are trained, validated, and tested." This was to be expected, as the course focused mainly on the technological methods of AI. On the other hand, items that covered content that did not occur in the course had low CSA gain values. Accordingly, the critical-appraisal factor had a lower overall learning gain because it included some items that dealt with the ethical, legal, and social aspects of AI, which were not covered in the evaluated course. Thus, our study provided initial evidence for an affirmative answer to RQ1. The adapted SNAIL questionnaire can be used to assess the learning gains of AI courses in a valid and reliable way.

Second, RQ2 asked whether AI literacy and attitudes toward AI were correlated, as this might be expected but would not be helpful for a criterion-valid assessment of learning gain. However, the learning gain scores of the three SNAIL factors correlated only very weakly with the two factors of the "Attitudes Toward Artificial Intelligence" scale. This could be an indication of discriminant validity because, in theory, AI literacy and attitudes toward AI are assumed to be two different constructs, representing cognitive/skill and affective aspects, respectively [9,16,20–22].

Third, RQ3 was asked to determine the extent to which participation in other AI courses (in addition to the AI course evaluated here) influenced learning gains. The use of other educational opportunities correlated significantly with retrospective and current self-assessment scores (especially on the TU factor), but not with learning gains. Accordingly, people who had already taken part in many AI courses (especially with a focus on technical understanding) tended to rate their AI literacy higher than did people who had comparatively little AI education. At the same time, however, the amount of actual learning (CSA gain) was unaffected by attending other AI courses. This made sense, because AI education before the course should already have had a positive influence on the retrospective assessment of one's own AI literacy, which in turn should have led to lower learning gains, due to a ceiling effect.

Furthermore, the reliability of the subscales of the adapted SNAIL also seemed to be satisfactory, as illustrated by the good-to-excellent internal consistency. Cronbach's $\alpha$ was high enough for the retrospective items, as well as for the post-assessment items, to justify the use of the adapted scale. In fact, the internal consistency of the scale was so satisfactory that one could consider removing some items to improve test efficiency. This would reduce the length of the questionnaire, which could increase participation rates, especially in the context of course evaluation.

The retrospective/post assessment seemed to yield valid and reliable results. However, it should once again be emphasized that the use of comparative self-assessment gains is particularly suitable for identifying between-subject differences, as well as differences between individual items [28,34]. If future research projects seek to apply adapted AI-literacy scales, it might, therefore, be advisable to calculate the comparative self-assessment gain rather than the traditional mean comparison via *t*-tests. If *t*-tests are nevertheless (additionally) conducted, the effect size, expressed, for example, by Cohen's d, should be included in any case. In this way, the strength of the learning effect can be estimated, at least in relative terms.

While the primary objective of this study was the validation of the SNAIL measurement instrument, a brief examination of the course itself and potential areas for improvement in course content was warranted. As previously noted, the course in question was not a general AI-literacy course, but focused on the technical aspects of AI. Consequently, most items that were subsumed under the technical-understanding factor yielded favorable

results. However, the findings also raised the question as to why course participants did not feel confident, for instance, in explaining the relationship between deep learning and machine learning (item TU2). Therefore, these aspects may merit heightened attention in future course iterations. Given this study's concentration on technical AI methods, our primary aim was not to enhance the learning outcomes regarding ethical items in the future. Nevertheless, course instructors may contemplate providing students with additional resources if they wish to further their knowledge in these areas or provide a broader AI-literacy course.

### 4.2. Limitations

As with any research, this study had some limitations. Even though students from different disciplines and backgrounds participated in the course, this study examined only one course (i.e., a single sample). In addition, some of the participants came from computer science backgrounds and, thus, they were both technically inclined and likely to have been familiar with some of the terminology. The original (un-adapted) SNAIL, however, was aimed at non-experts, i.e., individuals who had received little formal AI education. Furthermore, the selection of the SNAIL questionnaire was, although not completely without reason, relatively arbitrary. For reasons of evaluation efficiency, it was not possible to examine how respondents would have responded to other adapted questionnaires (e.g., [23–25]).

### 4.3. Future Research Directions

Future research projects should test the adapted SNAIL questionnaire in additional contexts and for larger courses. For example, the adapted scale should be used to evaluate courses in which the focus is not on technological AI aspects but on ethical, legal, and social features. The evaluation of AI teaching in specific disciplines would also be of interest, as promoted by Schleiss et al. [35]. For instance, whether the learning effects for medical students and engineering students differ for certain items could be investigated. In addition, it would be important to investigate whether complete novices would have produced a similar response scheme as that of the sample in this study. Furthermore, future evaluation studies should compare different AI-literacy-assessment tools to identify similarities and differences. Moreover, the relationship between AI literacy and attitudes toward AI should be further investigated in larger, preferably experimental, research projects to explore the causal direction of possible correlations. Finally, whether the number of items could be reduced without decreasing the internal consistency of the questionnaire should be examined, as this could increase testing efficiency. In addition to internal consistency, test–retest reliability could be investigated. This would be possible, for example, by carrying out a retention test some weeks or months after an AI course is completed.

### 5. Conclusions

This study presented preliminary evidence suggesting that even small adaptations of existing AI-literacy scales enables their use as AI-course-evaluation instruments. The combination of retrospective self-assessments on one's own competencies before starting a course and self-assessment after attending a course seems to lead to valid results, while simultaneously ensuring test efficiency. Overall, this study contributes to the development of valid, reliable, and efficient AI-course-evaluation instruments that allow a systematic assessment and improvement of AI education.

**Supplementary Materials:** The following supporting information can be downloaded at: https://www.mdpi.com/article/10.3390/educsci13100978/s1, Supplementary Material S1: Adaptation of the original questionnaire used in the study. Supplementary Material S2: Data Set.

**Author Contributions:** Conceptualization, M.C.L. and J.S.; methodology, M.C.L. and A.A.; formal analysis, M.C.L.; investigation, J.-O.P. and J.S.; resources, J.-O.P.; writing—original draft preparation, M.C.L.; writing—review and editing, J.S. and A.A.; visualization, M.C.L. and J.S.; project administration, J.-O.P. All authors have read and agreed to the published version of the manuscript.

**Funding:** This research received no external funding.

**Institutional Review Board Statement:** Not applicable.

**Informed Consent Statement:** Informed consent was obtained from all subjects involved in this study.

**Data Availability Statement:** The de-identified dataset on which the analyses in this study are based is available in Supplementary Material S2.

**Acknowledgments:** We want to express our appreciation to our supervisors Tobias Raupach and Sebastian Stober for their trust and support in implementing this project. Moreover, we also want to thank the reviewers for their feedback and constructive comments in the review process.

**Conflicts of Interest:** The authors declare no conflict of interest.

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
