# Peer review of "Evaluating AI Courses: A Valid and Reliable Instrument for Assessing Artificial-Intelligence Learning through Comparative Self-Assessment"

_education, doi:10.3390/educsci13100978_

Round 1

Reviewer 1 Report

Your paper is interesting and very clear.

I have three comments:

1. Regarding the title of Figure 2, you wrote "Mean values of retrospective and post self-assessments of all three factors." Did you mean the CSA gain? Also, the numbers' meaning needs to be clarified.

2. Do you find any differences between the groups? 

3. In the line 337 you wrote, "in theory AI literacy and attitudes towards AI are assumed to be two different constructs." You can add a reference or explain this statement.

Best regards.

The Quality of the English Language is good.

I just recommend being consistent in the use of terms: you use "retrospective" assessment (for example, in line 337) and, in other places, "then-assessment" (for example, in line 100).

Author Response

Comment 1: Regarding the title of Figure 2, you wrote "Mean values of retrospective and post self-assessments of all three factors." Did you mean the CSA gain? Also, the numbers' meaning needs to be clarified.

Response: Thank you very much for your helpful and valuable comments. Your assessment that we meant the CSA gain is correct. We have changed the title of Figure 2 according to your suggestion. We have also tried to define the meaning of the figures more clearly.

Comment 2: Do you find any differences between the groups?

Response: Unfortunately, it is not entirely clear to us what group differences you are referring to. The sample consisted of one group, and the creation of various sub-groups would only have been possible ex post. However, due to the relatively small overall sample, statistically significant differences between these ex post-groups (e.g., based on gender or study program) would not be particularly meaningful. For group comparisons, we usually aim for a minimum group size of n = 30 in order to be able to perform, for example, t-tests or ANOVAs. This would not have been possible based on our total sample.

Comment 3: In the line 337 you wrote, "in theory AI literacy and attitudes towards AI are assumed to be two different constructs." You can add a reference or explain this statement.

Response: We added references to papers defining AI literacy and attitudes towards AI, respectively. Furthermore, we explained that AI literacy is focused on cognition and skills, whereas attitudes towards AI are based on emotions, i.e., the “affective side” of human-AI-interaction (see line 346 ff.).

Comment 4: I just recommend being consistent in the use of terms: you use "retrospective" assessment (for example, in line 337) and, in other places, "then-assessment" (for example, in line 100).

Response: Thank you for this comment and suggestion. We have updated the respective parts to keep consistent terminology (i.e., using the term “retrospective”) throughout the manuscript.

Reviewer 2 Report

This is a well-constructed, valuable paper about the assessment of learning gains of AI literacy courses.  It is exploratory research and would be ideal for presentation at a conference.

The paper investigates the use of an existing AI literacy assessment tool for measuring AI learning gains. Although they mention different assessment tools, they choose the SNAIL scale developed by Laupichler et al (2022

Using a Delphi study, Laupichler et al (2022) identified the following aspects of AI literacy:  Technical understanding, Critical appraisal and Practical application.  These factors were subsequently used in the development of the SNAIL assessment tool. 

The main flaw in this article is the following:  The course content (as far as I could see) only addressed some of the aspects of AI literacy as defined by Laupichler et al (2022) and what is assessed in the SNAIL assessment tool.  The main focus of the course is to teach the basics of Neural Networks, which is but one aspect of AI literacy, according to Lauplicher et al. For this reason, SNAIL is not a good measurement tool for the efficacy of the course or perhaps the course is not a well-developed AI literacy course.  This of course raise a few questions: What does a typical AI literacy course entail then?  Is it even possible to have such a course given the different audiences?  Is it, therefore, possible to have one AI literacy assessment tool that will be applicable across all audiences and contexts?

Although the authors acknowledge these limitations, more can be done to strengthen the argument. Authors can map the course objectives to the AI literacy elements and give ideas on how their current course can be adapted to cater for the different aspects of AI literacy. 

Author Response

Response: Thank you for this helpful and valuable comment. We agree that the course was not a “pure” AI literacy course, since it was focused mainly on technical aspects of AI. Ethical aspects regarding AI, for instance, were not explicitly taught in the course.

While it may seem that the SNAIL questionnaire might not be the ideal tool for assessing the course, the specialization within the course actually proved its worth in demonstrating the questionnaire's effectiveness in measuring what it was designed to assess, thus establishing its validity. An interesting finding from this study was that the more technically-oriented items within the 'Technical Understanding' factor consistently showed a greater impact on learning outcomes compared to the ethical items within the 'Critical Appraisal' factor (refer to Figure 2). This observation raises an important point: if the measurement instrument had yielded a high learning effect across all items, it would have raised questions about why such a learning effect was observed when certain topics or concepts were not explicitly taught.

You rightly pointed out that our course may not align perfectly with what one might expect from a comprehensive AI literacy program. We acknowledge this observation to some extent. Our course, as previously explained, intentionally concentrated on specific facets of AI. Additionally, it attracted a technically inclined participant base. This alignment was deliberate because the course's primary objective was not to provide a holistic overview of AI but rather to delve deeply into its technical aspects. Consequently, we think that it was appropriate to include participants with a strong technical inclination.

It's important to clarify that the objective of our manuscript was never to present our course as an all-encompassing AI literacy program. Instead, our aim was to illustrate that the SNAIL questionnaire, as a measurement tool, is versatile enough to assess AI courses. We believe that the findings we present here can be extrapolated to other AI courses, especially those with a broader focus that encompass a wider spectrum of AI literacy topics. However, we acknowledge that a more thorough examination of these 'pure' AI literacy courses should be conducted in future studies.

Finally, we would like to address your comment regarding the mapping of course objectives and giving ideas on how their current course can be adapted to cater for the different aspects of AI literacy. While, as discussed above, this was not a primary aim of our research endeavor, we have nonetheless incorporated an additional paragraph in the discussion devoted to this matter. You can find this section in line 377 to 389. In addition, we have tried to make some of the points you have noted clearer at various points in the text (see passages marked in yellow).

Reviewer 3 Report

* In Lines 23 - 26, examples of AI are discussed; however, it may be useful to establish a definition of AI from which AI literacy is derived.

* The paragraph beginning at Line 106 is especially helpful to the reader in understanding the assessment used in the study.

* Line 199: Use the Title instead of the number. Readers have to scroll to the references to find it while they are reading.

* Supplementary material: Effective use of English/German translation

Author Response

Comment 1: In Lines 23 - 26, examples of AI are discussed; however, it may be useful to establish a definition of AI from which AI literacy is derived

Response: Thank you very much for sharing your helpful comments. While we generally agree that unknown constructs must be defined upon their first mention, this is unfortunately difficult for artificial intelligence, as there is no universally accepted definition for AI. However, we agree that the readership of the journal comes from very different disciplines, so it cannot be assumed that all people are familiar with the meaning of the term artificial intelligence. We have therefore added two relatively basic definitions, which are used in this or a similar form in most AI definitions.

Comment 2: The paragraph beginning at Line 106 is especially helpful to the reader in understanding the assessment used in the study

Response: Thank you!

Comment 3: Line 199: Use the Title instead of the number. Readers have to scroll to the references to find it while they are reading

Response: We have followed the suggestions and have added the title and the authors of the article to avoid readers having to scroll to the references.

Comment 4: Supplementary material: Effective use of English/German translation

Response: Thank you very much. We plan to publish the scales in both languages so that researchers in both language areas can use the SNAIL scales in their classrooms and research projects. Other languages may also be added in the future.

Round 2

Reviewer 2 Report

I believe my concerns have been addressed.